# HALO: A Unified Vision-Language-Action Model for Embodied Multimodal Chain-of-Thought Reasoning

**Quanxin Shou** [1][*]  **Fangqi Zhu** [1][*]  **Shawn Chen**  **Puxin Yan**  **Zhengyang Yan** [1]  **Yikun Miao** [1]  **Xiaoyi Pang** [1]
**Zicong Hong** [1]  **Ruikai Shi** [1]  **Hao Huang** [1]  **Jie Zhang** [1]  **Song Guo** [1]

{qshou,fangqi.zhu}@connect.ust.hk

## Abstract

Vision–Language–Action (VLA) models perform well in robotic manipulation, but often struggle in long-horizon or out-of-distribution scenarios due to the lack of explicit mechanisms for multimodal reasoning and anticipating how the world evolves under action. Recent works introduce textual chain-of-thought or visual subgoal prediction within VLA models, but still fail to offer a unified human-like framework for joint textual reasoning, visual foresight, and action prediction. To this end, we propose HALO, a unified VLA model that enables embodied multimodal chain-of-thought (EM-CoT) reasoning through textual reasoning, fine-grained visual subgoal prediction, and EM-CoT-augmented action prediction. We instantiate HALO with a Mixture-of-Transformers (MoT) architecture that decouples semantic reasoning, visual foresight, and action prediction into specialized experts with seamless cross-expert collaboration. To enable HALO learning at scale, we introduce an automated pipeline to synthesize EM-CoT training data along with a carefully crafted training recipe. Extensive experiments demonstrate that: (1) HALO achieves superior performance in both simulated and real world, surpassing baseline policy $\pi_0$ by 25.9% on the RoboTwin benchmark; (2) all proposed components of the training recipe and EM-CoT design help improve task success rate; and (3) HALO exhibits strong generalization under aggressive unseen environment randomization with our proposed EM-CoT reasoning.

---

[*]Equal contribution  [1]The Hong Kong University of Science and Technology, Hong Kong, China. Correspondence to: Song Guo <songguo@cse.ust.hk>.

*Proceedings of the 43rd International Conference on Machine Learning*, Seoul, South Korea. PMLR 306, 2026. Copyright 2026 by the author(s).

## 1. Introduction

Vision-Language-Action (VLA) models (Black et al., 2024; NVIDIA et al., 2025) have shown a compelling path toward general-purpose robotic manipulation. However, most existing VLAs map perceptual inputs directly to motor commands, lacking explicit mechanisms for reasoning about task structure or anticipating how the environment will evolve under motion and contact. This limitation becomes particularly pronounced in long-horizon or out-of-distribution scenarios—such as novel layouts, unfamiliar objects, or contact-rich interactions—where successful execution depends more on deliberation and foresight than on reactive pattern matching.

Recent work has sought to address this limitation by introducing intermediate reasoning processes like human. Inspired by the success of Chain-of-Thought reasoning (Wei et al., 2023) in large language models, Zawalski et al. (2025) train VLAs to perform multi-step reasoning prior to action prediction, enabling task decomposition but without grounding such reasoning in explicit visual state evolution. In contrast, Zhao et al. (2025) autoregressively generate subgoal images as visual intermediate steps, but lack semantic reasoning capabilities. Achieving human-like textual reasoning and visual imagination capability within a unified framework remains challenging, as it demands strong multimodal generation while preserving reasoning capacity. Existing approaches (Zhao et al., 2025; Gu et al., 2025) often tightly couple visual generation with VLMs via discrete visual token prediction, which may hinder rich understanding capabilities of VLMs. Consequently, a unified architecture that jointly supports multimodal reasoning, visual generation, and action prediction remains an open problem.

To address this, we propose HALO, a unified VLA model that enables embodied multimodal chain-of-thought (EM-CoT) reasoning. As illustrated in Figure 1, HALO first performs textual reasoning and task planning, then predicts subgoal images to provide fine-grained visual guidance, and finally uses the resulting EM-CoT as context for action prediction. This design is specifically engineered to emulate the human cognitive process of deliberate reasoning,

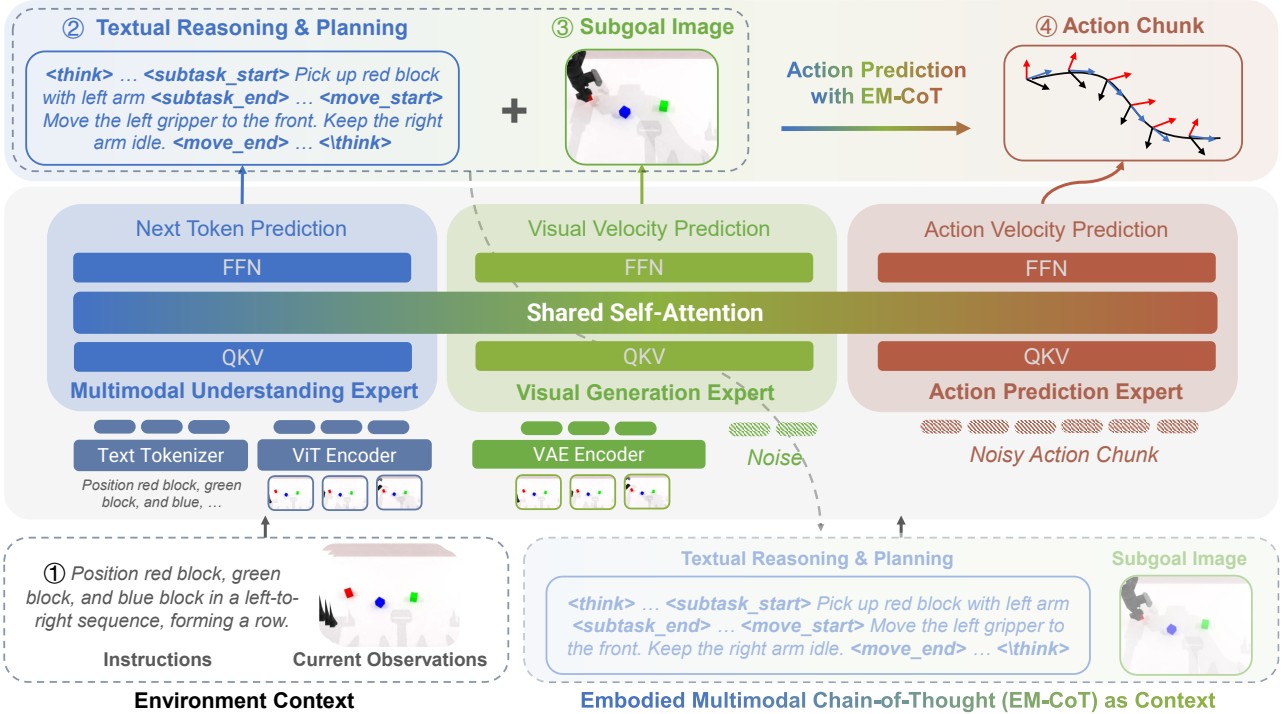

Figure 1. HALO first performs textual reasoning and task planning, then predicts visual subgoals for fine-grained guidance, and finally generates actions conditioned on EM-CoT. This process is implemented using a Mixture-of-Transformers (MoT) architecture that integrates a multimodal understanding expert, a visual generation expert, and an action prediction expert through shared self-attention.

mental imagination, and subsequent execution, rendering HALO particularly well-suited for tackling complex, long-horizon tasks that require intricate sequential reasoning and sustained contextual understanding over extended periods.

HALO incorporates several key innovations to realize this vision. First, to natively support embodied multimodal reasoning, HALO adopts a Mixture-of-Transformers (MoT) architecture (Liang et al., 2024) that decouples textual reasoning, visual foresight, and action prediction into three specialized experts. These experts collaborate seamlessly through shared self-attention, while preserving each expert's natural generative workflow—autoregressive text generation for reasoning and diffusion-based prediction for visual states and actions—thereby avoiding potential conflicts that arise when heterogeneous capabilities are forced into a single monolithic model. Second, we introduce an automatic EM-CoT data synthesis pipeline built upon action primitives and VLM annotator. The pipeline performs task planning first and then decomposes long-horizon robot trajectories into a sequence of sub-tasks, each paired with a corresponding subgoal image and explicit reasoning process. This enables automatic and scalable embodied multimodal reasoning supervision for our unified VLA model. Third, we propose a carefully designed training recipe that combines broad generalization with embodied reasoning specialization. (1) In a versatile pre-training stage, HALO is trained on diverse care-

fully crafted data sources, including general VQA datasets for robust multimodal understanding, egocentric and robotic video datasets for learning contact-rich physical dynamics, and heterogeneous robot trajectory datasets for acquiring foundational manipulation skills. (2) In a subsequent EM-CoT–augmented fine-tuning stage, the model is co-trained on EM-CoT and VQA data to inject EM-CoT reasoning capability while preserving its general world knowledge.

We conduct extensive experiments in both the RoboTwin 2.0 simulator (Chen et al., 2025) and real-world environments to validate the effectiveness of HALO. The results demonstrate that HALO achieves consistently stronger overall performance across both simulated and real-world settings, establishing a new state-of-the-art with average success rates of 72.3% and 27.1% on the Easy and Hard settings of RoboTwin, surpassing $\pi_0$ by 25.9 and 10.8 percentage points, respectively. In addition, we perform comprehensive ablation studies to verify: (1) the effectiveness of the proposed EM-CoT, showing that combining textual reasoning with visual foresight is especially beneficial under challenging domain-randomized settings; and (2) the importance of each category of pre-training tasks, all of which provide measurable benefits to downstream performance. We further demonstrate the strong generalization capability of HALO in both simulation and real robots, and qualitatively show that it can generate reasonable EM-CoT to guide action pre-

diction in unseen scenarios. Taken together, these results highlight HALO as a scalable and generalizable paradigm for human-like VLA reasoning.

## 2. Related Work

**Vision-Language-Action Models.** Most VLA models are built on top of Vision-Language Models (VLMs) pre-trained on large-scale web data, inheriting rich multimodal world knowledge. Representative architectures such as Open-VLA (O'Neill et al., 2024) and FAST (Pertsch et al., 2025) directly extend pre-trained VLMs by introducing discrete action tokens and predicting actions autoregressively to solve embodied tasks. The $\pi_0$ series (Black et al., 2024; Intelligence et al., 2025) further introduce a separate action expert along with the pre-trained VLM for flow-based action prediction, achieving better performance and efficiency.

Beyond equipping VLM with only action prediction capability, unified VLA architectures (Bi et al., 2025; Wang et al., 2025; Cen et al., 2025; Lv et al., 2025) have been proposed to enhance dynamic awareness by jointly modeling textual understanding, visual generation, and action prediction. These methods incorporate image or video generation components to predict future environmental states, using either unified (Cen et al., 2025; Wang et al., 2025) or mixed (Bi et al., 2025; Lv et al., 2025) architectures that are trained together with action prediction to capture underlying environment dynamics. However, these methods lack explicit textual or visual reasoning capability, limiting their performance in complex and dynamic environments.

**Textual and Visual Reasoning for Robotic Intelligence.** Reasoning is a crucial step towards robotic intelligence (Le-Cun, 2022). Motivated by the success of Chain-of-Thought reasoning in language models, recent work extends explicit reasoning mechanisms to embodied decision making. MoTVLA (Bi et al., 2025) proposes a textual reasoning framework that performs planning and reasoning prior to action execution. UP-VLA (Zhang et al., 2025), CoTVLA (Zhao et al., 2025) and InternVLA-A1 (Cai et al., 2026) further introduce visual reasoning by generating goal images and videos to guide downstream action prediction.

More recently, ManualVLA (Gu et al., 2025) tries to combine visual and textual reasoning through a shared planning expert that produces both textual instructions and visual subgoals. However, this design couples image and text generation within a single autoregressive model, which may degrade the reasoning capability of the underlying VLM. In contrast, our proposed **embodied multimodal chain-of-thought (EM-CoT)** employs a **Mixture-of-Transformers (MoT) architecture** for joint textual reasoning, visual generation, and action prediction. Our decoupled design enables more effective reasoning across visual and textual modali-

ties before executing actions.

## 3. Method

In this section, we delineate the core design principles of HALO, focusing on its unified architecture, the embodied multimodal chain-of-thought (EM-CoT) data pipeline, and the associated training recipe. Collectively, these components empower the model to engage in textual reasoning, visual foresight, and grounded action prediction.

### 3.1. Problem Formulation

We formulate robotic manipulation as a sequential decision-making problem. Let $\tau = \{(\mathbf{o}_t, \mathbf{l}, \mathbf{a}_t)\}_{t=1}^T$ denote a trajectory, comprising visual observations $\mathbf{o}_t \in \mathcal{O}$, language instructions $\mathbf{l} \in \mathcal{L}$, and continuous actions $\mathbf{a}_t \in \mathcal{A}$. Traditional VLA models typically learn a monolithic policy $\pi_\theta(\mathbf{a}_{t:t+m} \mid \mathbf{l}, \mathbf{o}_{t-k:t})$ that directly maps history observations and instructions to action chunks. Such purely reactive policies often suffer from performance degradation when facing long-horizon or complex manipulation tasks due to a lack of intermediate reasoning.

In this case, we aim to integrate explicit EM-CoT as a robust conditioning mechanism in HALO and make its generation process align with the typical human cognitive pathway of "thinking-imagination-execution". To achieve our goal, we decompose the generation process of HALO into three distinct phases: textual reasoning, visual foresight, and action prediction. Then the problem is transformed into learning a unified policy $\pi_\theta$ that jointly models the following mappings:

$$\mathbf{r} \sim P_\theta(\cdot \mid \mathbf{l}, \mathbf{o}_{t-k:t}) \tag{1}$$

$$\hat{\mathbf{o}}_{t+h} \sim P_\theta(\cdot \mid \mathbf{l}, \mathbf{o}_{t-k:t}, \mathbf{r}) \tag{2}$$

$$\mathbf{a}_{t:t+m} \sim \pi_\theta(\cdot \mid \mathbf{l}, \mathbf{o}_{t-k:t}, \mathbf{r}, \hat{\mathbf{o}}_{t+h}) \tag{3}$$

where $\mathbf{r}$ represents the generated textual chain-of-thought and $\hat{\mathbf{o}}_{t+h}$ denotes the predicted visual subgoal. Under this formulation, the action chunk $\mathbf{a}_{t:t+m}$ is not a simple reaction to inputs, but the result of a deliberate reasoning trace and physical world modeling, ensuring that every control command is strategically planned and visually grounded.

### 3.2. Unified Architecture

A unified architecture is adopted in HALO, as illustrated in Figure 1. Drawing inspiration from the capability of BAGEL (Deng et al., 2025) to harmonize multimodal tasks, we adopt a Mixture-of-Transformers (MoT) (Liang et al., 2024) architecture comprising three specialized experts: *Multimodal Understanding*, *Visual Generation*, and *Action Prediction*. These three experts specialize in distinct modalities and are responsible for textual reasoning, visual generation, and action prediction, respectively. While maintaining

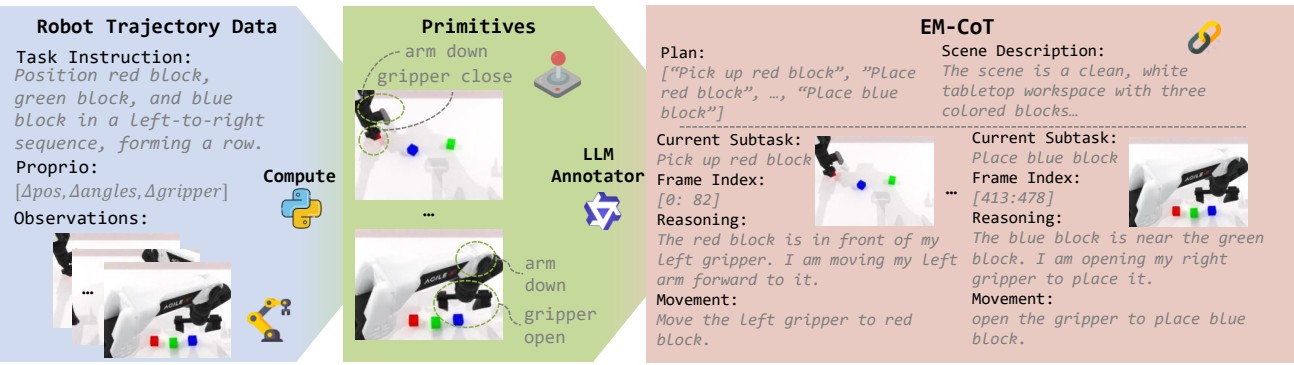

*Figure 2.* Overview of EM-CoT Data Synthesis Pipeline. The pipeline converts raw robotic trajectories into EM-CoT data in three phases: (1) action primitives are extracted from robot proprioception via rule-based matching; (2) a VLM acts as an annotator to generate task plans, decompose trajectories into subtasks, and align each subtask with explicit textual reasoning; and (3) the terminal frame of each subtask is selected as a visual subgoal image, producing structured embodied multimodal chain-of-thought supervision.

independent parameter sets, they interact through a shared self-attention mechanism to enable rich cross-modal interactions. With such an architecture, HALO operates as a unified VLA model that simultaneously possesses the capability to generate detailed textual reasoning and to produce explanatory subgoal images.

The switching mechanism between modalities is explicitly controlled via special tokens. By default, the model operates as an auto-regressive planner; however, the generation of specific tokens (e.g., ⟨visual_start⟩ or ⟨action_start⟩) triggers the routing of hidden states to the visual generation expert or the action prediction expert, respectively. In order to manage the information flow among experts, we implement a carefully structured masking strategy within the shared self-attention. As illustrated in Figure 3, a causal mask is applied to textual tokens to enforce autoregressive generation. Conversely, visual tokens employ bidirectional attention within each frame to capture global spatial dependencies, while maintaining causal masking for interactions across frames or modalities. Crucially, to prevent information leakage, noise tokens are restricted from attending to their corresponding ground-truth targets. Furthermore, all other tokens are masked from attending to noise tokens, ensuring they only aggregate information from valid non-noise contexts.

To facilitate training stability, we initialize each expert using the Qwen2.5-1.5B LLM (Yang et al., 2025), yielding a cumulative parameter count of approximately 4.5B. To project heterogeneous data into a unified representation space, we employ modality-specific encoders. For *text*, we adopt the standard Qwen2.5 tokenizer (Bai et al., 2023), expanding the vocabulary with special control tokens—specifically ⟨think_start⟩, ⟨vision_start⟩, and ⟨action_start⟩, among others—to strictly demarcate the textual reasoning, visual generation, and acting phases, thereby enforcing a structured and interpretable workflow. In the *vision* domain, we decouple the encoding mechanisms for semantic under-

standing and visual generation to address their distinct functional requirements. For the *understanding* task, we employ a ViT (Dosovitskiy, 2020) initialized with SigLIP2 (Tschannen et al., 2025) to leverage robust semantic features. To accommodate diverse real-world inputs without geometric distortion, this encoder is further enhanced with NaViT (Dehghani et al., 2023), enabling the processing of images in their native aspect ratios. For the *visual generation* task, we utilize a pre-trained VAE (Kingma & Welling, 2013; Labs et al., 2025) optimized for high-fidelity image compression and reconstruction. Finally, for the *action* modality, given its low-dimensional and physically structured nature, a linear projection suffices to map actions into the model's hidden dimension while preserving intrinsic structural information. See Appendix A for further architectural details.

### 3.3. EM-CoT Data Pipeline

To support the generation of intermediate textual reasoning trace **r** and visual subgoal $\hat{\mathbf{o}}_{t+h}$, we construct an automatic pipeline to synthesize EM-CoT data from raw robot trajectories at scale. As shown in Figure 2, the pipeline operates in three phases. Firstly, we translate continuous low-level actions into high-level motion primitives via rule-based matching following Belkhale et al. (2024). Secondly, we utilize a large-scale Vision-Language Model (Qwen3-VL (Bai et al., 2025)) to augment the trajectory with dense textual reasoning, including task narratives and subtask decomposition. Finally, the terminal frame of each subtask is designated as its corresponding visual subgoal, providing a sparse supervision signal that effectively lowers the learning difficulty. This process yields a high-quality dataset $\mathcal{D}_{\text{ft}}$ where every trajectory is augmented with aligned textual reasoning **r** and visual goals $\hat{\mathbf{o}}$, enabling the supervision of the intermediate steps defined in Eq. (1)-(2). See Appendix B for more details on this pipeline, including the pseudo code and tailored prompt used in each phase.

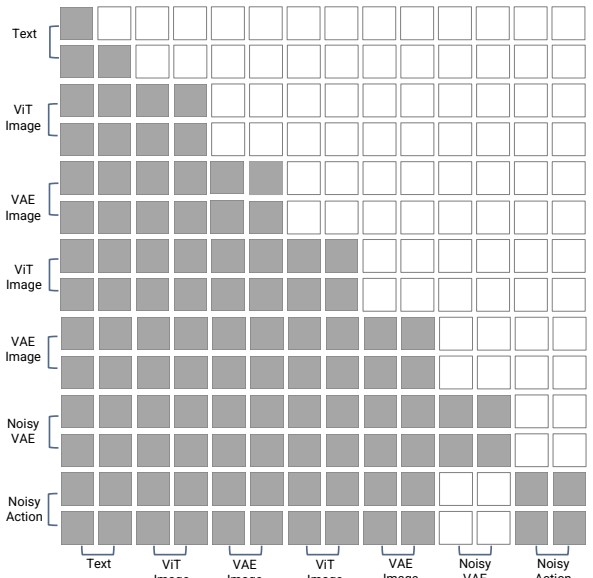

*Figure 3.* Attention Masking Strategy for EM-CoT. The mechanism employs distinct attention patterns for different token types: (1) Spatial tokens (VAE-encoded) and semantic tokens (ViT-encoded) utilize bidirectional attention within their respective frames to model local spatial context. (2) Noise tokens corresponding to a specific frame or action chunk also attend bidirectionally to one another. (3) All cross-frame and cross-modality interactions, as well as textual token generation, strictly adhere to causal attention constraints. (4) To prevent leakage, non-noise tokens are masked from attending to noise tokens.

### 3.4. Training Recipe

To evolve HALO into a capable deliberative unified VLA, we employ a two-stage training paradigm: (1) *Versatile Pre-training*, which establishes a robust generalist foundation, and (2) *EM-CoT-Augmented Fine-tuning*, which elicits structured multimodal reasoning capabilities.

**Stage 1: Versatile Pre-training.** The primary objective of this phase is to unify multimodal understanding, physical dynamics prediction, and foundational manipulation skills into a single architecture. To this end, we curate a massive, heterogeneous dataset spanning three distinct domains—Visual Question Answering (VQA), Visual Generation (VG), and Action Prediction (AP)—as illustrated in Fig. 4. This diversity ensures the model develops a generalized representation capable of supporting complex downstream reasoning.

- VQA (Multimodal understanding): We utilize LLaVA-NeXT-779k (Liu et al., 2024a) to provide high-quality multimodal conversational priors. This task aligns linguistic instructions with visual contexts via a standard cross-entropy loss, denoted as $\mathcal{L}_{CE}$.
- VG (Visual Generation): To instill physical common sense, we integrate robotic trajectories from OXE (O'Neill

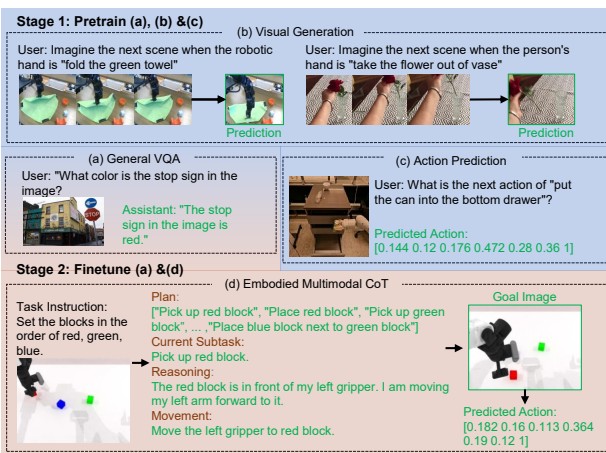

*Figure 4.* Overview of dataset recipe. HALO is trained in two stages. Stage 1 pre-trains the model on (a) general VQA, (b) visual generation, and (c) action prediction to support multimodal reasoning, future state foresight, and action prediction. Stage 2 retains (a) general VQA and introduces (d) EM-CoT–augmented fine-tuning for embodied multimodal reasoning.

et al., 2024) with ego-centric manipulation videos from SSv2 (Goyal et al., 2017). We formulate visual foresight as a future frame prediction task, $(l, I_{t-k:t}) \rightarrow I_{t+h}$, trained using a flow-matching MSE loss, $\mathcal{L}_{MSE}$. Crucially, we employ a dual-path visual pathway that integrates complementary semantic and spatial representations: a ViT branch first captures high-level semantic context, while a VAE branch provides spatially grounded latent features that are compact and generation-friendly. By combining these two pathways, the model jointly exploits semantic dynamics and spatial structure for prediction.

- AP (Imitation Learning): We repurpose the OXE dataset for direct action prediction. The model learns to predict the next action chunk $a_t$ given the history, supervised by an $L_1$ flow-matching loss, $\mathcal{L}_{L1}$.

To address the varying optimization difficulties across these tasks, we balance the pre-training objective by assigning higher weights to manipulation intensive tasks. The total pre-training loss is defined as:

$$\mathcal{L}_{pt} = 0.25\mathcal{L}_{CE} + 0.5\mathcal{L}_{MSE} + \mathcal{L}_{L1}. \qquad (4)$$

**Stage 2: EM-CoT-Augmented Fine-tuning.** Building upon the pre-trained foundation, we transition HALO from a reactive policy to a deliberative one by fine-tuning it on our EM-CoT dataset, $\mathcal{D}_{ft}$ (see Sec. 3.3). This stage augments standard robotic trajectories with explicit textual reasoning traces **r** and visual subgoals **ô**.

Adhering to the sequential structure defined in Eq. (1)–(3), the model is trained to orchestrate a complete "thought-foresight-action" chain. The fine-tuning objective mini-

*Table 1.* Quantitative results in simulation on RoboTwin 2.0. Each task is evaluated 100 times under the Easy (Clean) and Hard (Domain-randomized) settings, and the average success rate across 50 tasks is reported. The complete results are provided in Appendix D.

| Task | $\pi_0$ | | RDT-1B | | Diffusion Policy | | HALO-w/o EM-CoT | | HALO | |
|---|---|---|---|---|---|---|---|---|---|---|
| | Easy | Hard | Easy | Hard | Easy | Hard | Easy | Hard | Easy | Hard |
| Adjust Bottle | 90 | 56 | 81 | 75 | 97 | 0 | 100 | 11 | 96 | 8 |
| Beat Block Hammer | 43 | 21 | 77 | 37 | 42 | 0 | 73 | 7 | 91 | 4 |
| Blocks Ranking RGB | 19 | 5 | 3 | 0 | 0 | 0 | 71 | 5 | 77 | 8 |
| Blocks Ranking Size | 7 | 1 | 0 | 0 | 1 | 0 | 29 | 2 | 43 | 1 |
| Click Alarmclock | 63 | 11 | 61 | 12 | 61 | 5 | 97 | 6 | 95 | 17 |
| Click Bell | 44 | 3 | 80 | 9 | 54 | 0 | 100 | 26 | 100 | 16 |
| Dump Bin Bigbin | 83 | 24 | 64 | 32 | 49 | 0 | 85 | 39 | 89 | 25 |
| Grab Roller | 96 | 80 | 74 | 43 | 98 | 0 | 99 | 77 | 94 | 46 |
| · · · | | | | | | | | | | |
| Shake Bottle Horizontally | 99 | 51 | 84 | 51 | 59 | 18 | 99 | 37 | 99 | 76 |
| Shake Bottle | 97 | 60 | 74 | 45 | 65 | 8 | 99 | 38 | 99 | 75 |
| Stack Blocks Three | 17 | 0 | 2 | 0 | 0 | 0 | 79 | 2 | 66 | 43 |
| Stack Blocks Two | 42 | 1 | 21 | 2 | 7 | 0 | 93 | 12 | 90 | 55 |
| Stack Bowls Three | 66 | 24 | 51 | 17 | 63 | 0 | 66 | 17 | 73 | 30 |
| Stack Bowls Two | 91 | 41 | 76 | 30 | 61 | 0 | 95 | 27 | 94 | 57 |
| Stamp Seal | 3 | 4 | 1 | 0 | 2 | 0 | 30 | 3 | 50 | 19 |
| Turn Switch | 27 | 23 | 35 | 15 | 36 | 1 | 51 | 7 | 54 | 27 |
| **Average** | 46.4 | 16.3 | 34.5 | 13.7 | 28.0 | 0.6 | 70.0 | 21.1 | **72.3** | **27.1** |

mizes the joint loss:

$$\mathcal{L}_{\text{ft}} = \mathcal{L}_{\mathbf{r}} + \mathcal{L}_{\hat{\mathbf{o}}} + \mathcal{L}_{\mathbf{a}}, \qquad (5)$$

where $\mathcal{L}_{\mathbf{r}}$, $\mathcal{L}_{\hat{\mathbf{o}}}$, and $\mathcal{L}_{\mathbf{a}}$ represent the losses for textual reasoning, visual subgoal generation, and action prediction, respectively. To mitigate catastrophic forgetting of general multimodal capabilities during this specialization, we incorporate general VQA data into the fine-tuning corpus (Cheang et al., 2025). This process results in a model that effectively bridges high-level semantic intent with precise, visually grounded motor control for long-horizon manipulation. Please refer to Appendix C for implementation details.

## 4. Experiments

We conduct extensive experiments to evaluate the effectiveness of HALO equipped with EM-CoT. Our study focuses on: (i) whether the proposed unified VLA architecture with EM-CoT improves overall performance and generalization; (ii) whether HALO can generate informative EM-CoT reasoning, particularly under novel settings; (iii) whether multimodal reasoning outperforms using only textual reasoning or visual subgoal prediction, and how the proposed training recipe contributes to performance; and (iv) whether HALO demonstrates superior performance in real-world settings.

### 4.1. Experiment Settings

During pre-training, following Bagel (Deng et al., 2025), training samples are concatenated to a maximum sequence length of 27k tokens, and Flex Attention (Dong et al., 2024) is adopted to accelerate training; HALO is trained for 90k

steps at this stage. For fine-tuning, we use raw trajectories collected from RoboTwin 2.0 (Chen et al., 2025) and real-world. The simulation dataset contains 2,500 expert demonstrations (50 per task) collected in clean environments, while the real-world dataset consists of 320 demonstrations (80 per task). With these data, HALO is fine-tuned for 110k steps in simulation and 80k steps in real world. During inference, task instructions together with $c = 3$ observation frames are provided as input. We set the action chunk length to $K = 16$ for simulation and $K = 50$ for real-world.

### 4.2. Simulation Results

We conduct simulation experiments on RoboTwin 2.0 (Chen et al., 2025), a comprehensive benchmark comprising 50 challenging manipulation tasks. We compare HALO with several competitive baselines, including $\pi_0$ (Black et al., 2024), RDT (Liu et al., 2024b), and Diffusion Policy (Chi et al., 2025). Baseline results are taken directly from the official RoboTwin 2.0 leaderboard.[1] To isolate the contribution of EM-CoT, we further include a variant of our method without EM-CoT, denoted as HALO-w/o EM-CoT.

Table 1 shows the performance of HALO and all baselines on RoboTwin 2.0. It can be observed that HALO consistently outperforms all competitive baselines across both Easy and Hard settings. The average success rate of HALO reaches 72.3% on Easy tasks and 27.1% on Hard tasks, surpassing baseline policy $\pi_0$ by 25.9 and 10.8 percentage points, respectively. Particularly, the substantial relative perfor-

---

[1] https://robotwin-platform.github.io/leaderboard

*Table 2.* Ablation studies on training recipe and EM-CoT. Panel A: training recipe ablation (V/T/A denote visual generation, textual VQA, and action prediction data. Note that in the *Full* setting, we pre-train the model with V&T&A datasets and then fine-tune it on the official RoboTwin 2.0 dataset). Panel B: EM-CoT ablation (V = visual subgoal images, T = textual reasoning).

| Panel A: Training Recipe | | | Panel B: EM-CoT | | |
|---|---|---|---|---|---|
| Setting | Easy | Hard | Setting | Easy | Hard |
| Full | **70.0** | **21.1** | HALO | **72.3** | **27.1** |
| w/o V | 58.2 | 10.5 | w/o T | 71.7 | 18.3 |
| w/o V+T | 42.9 | 3.9 | w/o V | 70.3 | 22.5 |
| w/o V+T+A | 32.4 | 0.0 | w/o V & T | 70.0 | 21.1 |

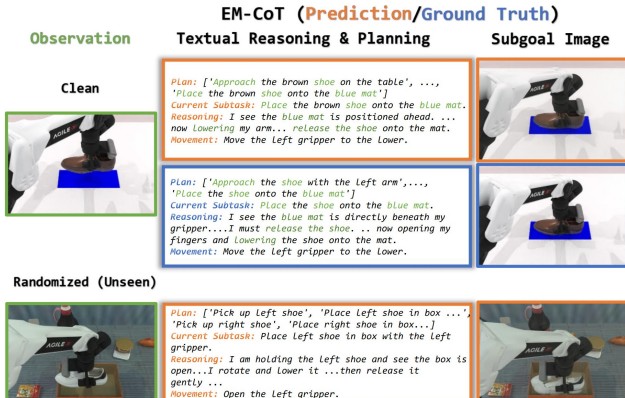

*Figure 5.* Qualitative Analysis of the EM-CoT. We highlight (i) accurate textual reasoning and subgoal image generation in the clean setting, and (ii) robust EM-CoT generalization under the unseen domain-randomized settings.

mance gaps (i.e., 55.8% and 66.3%) between HALO and $\pi_0$, especially on Hard tasks, indicate that HALO can handle out-of-distribution (OOD) challenges more robustly. This stems from the fact that HALO not only possesses robust foundational capabilities but also features EM-CoT capabilities meticulously tailored for intricate, complex scenarios. Besides, even without the explicit reasoning chain, HALO-w/o EM-CoT already surpasses the strongest baseline ($\pi_0$) by substantial margins of 23.6 and 4.8 percentage points on Easy and Hard tasks, demonstrating the powerful general foundation established by our versatile pre-training.

In contrast, traditional reactive policies such as Diffusion Policy struggle significantly in randomized environments, with success rates dropping to near-zero (0.6%) in Hard settings. While $\pi_0$ and RDT-1B exhibit some degree of robustness, they plateau at significantly lower performance levels, whereas HALO continues to improve as it incorporates more structured reasoning. On fine-grained tasks such as "Blocks Ranking Size" and "Stamp Seal," HALO achieves a multi-fold increase in success rates compared to baselines, demonstrating its ability to manage intricate object interactions. These results underscore the effectiveness and robustness of HALO for complex robotic manipulation.

## 4.3. Ablation Study

We perform ablation studies to validate the effectiveness of HALO's mechanism design, including the versatile pre-training and the EM-CoT-augmented Fine-tuning. The results are shown in Table 2.

*Effectiveness of the versatile pre-training.* As depicted in Panel A, the full versatile pre-training utilizing all three diverse datasets achieves the best performance, reaching 70.0%/21.1% on Easy/Hard tasks. Conversely, even the selective removal of a single data modality, such as visual generation data (w/o V), leads to a significant degradation in performance, particularly on hard tasks. Further progressive exclusion of textual (w/o V+T) and action prediction data (w/o V+T+A) results in a continuously worsening performance trajectory. Notably, without any pre-training (w/o V+T+A), the model's performance falls to a complete 0% on hard tasks, demonstrating that pre-training is an absolutely foundational requirement for establishing the core competencies necessary to tackle complex problems. Collectively, these findings underscore the indispensable and complementary roles of the versatile pre-training step.

*Effectiveness of EM-CoT.* As shown in Panel B, the full EM-CoT design achieves 72.3%/27.1% on Easy/Hard tasks, outperforming variants without textual reasoning (71.7%/18.3%), visual subgoals (70.3%/22.5%), or both components (70.0%/21.1%). The gain is particularly pronounced on Hard tasks, where multimodal reasoning improves robustness under domain randomization. These results validate the effectiveness of our proposed EM-CoT.

## 4.4. Qualitative Results of EM-CoT

To qualitatively assess the efficacy of EM-CoT, we visualize representative rollouts from both the clean and randomized settings of RoboTwin 2.0, as illustrated in Figure 5.

In the clean setting, EM-CoT demonstrates high fidelity, with predicted textual reasoning and visual subgoals closely aligning with the ground truth. The result indicates that HALO effectively masters the underlying task logic, providing a clear, self-generated multimodal roadmap that anchors final action in a semantically and visually consistent manner.

Under aggressive environmental randomization, EM-CoT still exhibits significant robustness while maintaining performance despite substantial visual shifts. As shown in the last row of Figure 5, HALO correctly identifies targets and plans trajectories even when object appearances and backgrounds significantly differ from the training distribution. This suggests our EM-CoT paradigm allows the agent to perform true semantic reasoning rather than relying on simple pattern matching. By enforcing a deliberate reasoning process, the model effectively performs low-level control guided by high-level reasoning under the challenging out-

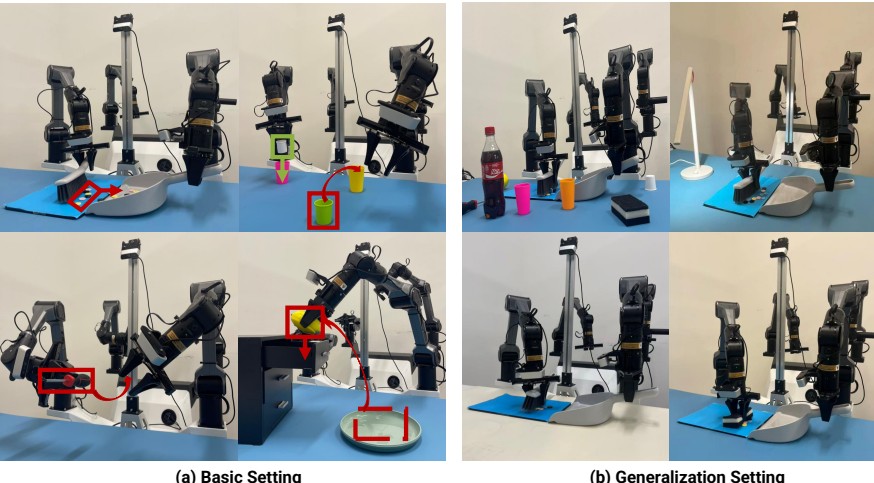

**(a) Basic Setting**          **(b) Generalization Setting**

*Figure 6.* Real-World Task Settings. (a) Basic Setting: Four tasks, including *tool-use sweeping*, *bimanual cup nesting*, *inter-arm screwdriver handover*, and *placing an object into a drawer*. (b) Generalization Setting: Challenging scenarios involving visual distractions (e.g., Coke bottles and cups), lighting variations, background variations (e.g., changing tablecloth colors), and novel objects (e.g., replacing the broom with a sponge).

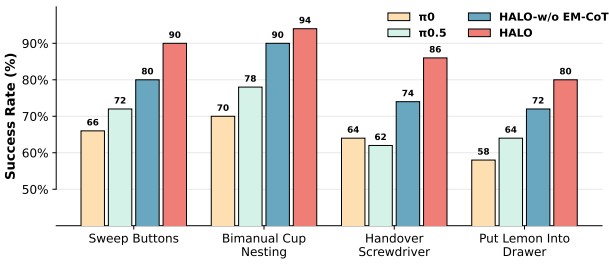

*(a)* The results of basic setting.

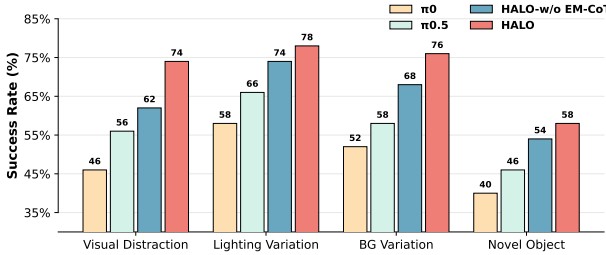

*(b)* The results of generalization setting.

*Figure 7.* Quantitative results of real-world experiments on both the basic setting and the generalization setting. Each task is evaluated over 50 trials, and the success rate is reported.

of-distribution scenarios.

### 4.5. Real-World Results

We further evaluate HALO on a real-world Cobot Mobile ALOHA platform on four long-horizon manipulation tasks, including *tool-use sweeping*, *bimanual cup nesting*, *inter-arm screwdriver handover*, and *placing an object into a drawer* (Figure 6). These tasks require multi-step planning, semantic grounding, and coordinated dual-arm control skills,

making them valuable for evaluating the effectiveness of EM-CoT reasoning in real-world settings.

As shown in Figure 7, HALO consistently outperforms the baseline policies $\pi_0$ and $\pi_{0.5}$ under both the basic and generalization settings. While the baselines suffer noticeable performance degradation in the presence of visual distractions, lighting and background variations, and novel objects, HALO remains robust and achieves the highest success rates across all tasks. These results demonstrate that EM-CoT reasoning enables more stable long-horizon execution and stronger generalization in real-world robotic manipulation.

## 5. Conclusion

In this paper, we propose HALO, a unified VLA model that enables embodied multimodal chain-of-thought (EM-CoT) reasoning through joint textual reasoning, visual foresight, and action prediction using a Mixture-of-Transformers (MoT) architecture. We also introduce an automated pipeline for synthesizing EM-CoT training data, along with a carefully crafted training recipe. Extensive experiments and ablation studies validate the effectiveness of HALO in the challenging out-of-distribution and real-world settings, highlighting HALO as a scalable and generalizable human-like VLA reasoning paradigm.

## Impact Statement

This paper aims to advance the field of robotic intelligence, with potential applications in industrial robotic systems. We do not identify any specific or unique societal risks arising directly from this work beyond those commonly associated with robotics research.

## Acknowledgement

This research was supported by funding from the Hong Kong RGC General Research Fund (152228/23E, 162161/24E, 162116/25E, 162180/25E), the National Natural Science Foundation of China (NSFC) Key Program (No. 62532005), the Collaborative Research Fund (No. C1042-23GF, No. C5097-25G), the NSFC/RGC Collaborative Research Scheme (Grant No. 62461160332 and CRS_HKUST602/24), the Research Impact Fund (No. R5011-23F), the Areas of Excellence Scheme (AoE/E-601/22-R), and InnoHK (HKGAI).

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

# A. Details on Model Architecture

## A.1. LLM Backbone

HALO leverages Qwen-2.5-1.5B LLM (Yang et al., 2025) as initial backbone. We now illustrate its configuration in Table 3.

*Table 3.* Qwen2.5-1.5B-Instruct Model Configuration Parameters

| Parameter | Value | Parameter | Value |
|---|---|---|---|
| architectures | ["Qwen2ForCausalLM"] | num_attention_heads | 12 |
| attention_dropout | 0.0 | num_hidden_layers | 28 |
| bos_token_id | 151643 | num_key_value_heads | 2 |
| eos_token_id | 151645 | rms_norm_eps | $1 \times 10^{-6}$ |
| hidden_act | "silu" | rope_scaling | null |
| hidden_size | 1536 | rope_theta | 1000000.0 |
| initializer_range | 0.02 | sliding_window | 32768 |
| intermediate_size | 8960 | tie_word_embeddings | true |
| max_position_embeddings | 32768 | torch_dtype | float32 |
| max_window_layers | 21 | transformers_version | 4.49.0 |
| model_type | "qwen2" | use_cache | true |
| vocab_size | 151933 | use_sliding_window | false |

## A.2. Multimodal Encoders

To jointly train all three modalities, we implement different encoders to map multimodal inputs into tokens, stated as follows:

- Textual Input: We utilize Qwen2.5's original tokenizer (Bai et al., 2023).
- Visual Input: We employ separate visual encoders to handle visual understanding and generation independently. For visual understanding, we utilize a ViT (Dosovitskiy, 2020) encoder initialized with SigLIP2-so400m/14 (Tschannen et al., 2025) (pre-trained at 384×384). To enhance high-resolution perception, we interpolate position embeddings to support inputs up to 980×980 and integrate NaViT (Dehghani et al., 2023) to process images in their native aspect ratios. A two-layer MLP projector is adopted to align visual features with LLM's embedding space. For visual generation, we leverage the pre-trained VAE (Kingma & Welling, 2013) from FLUX (Labs et al., 2025), featuring a downsample ratio of 8 and a latent channel of 16. The VAE latents are processed by flattening 2×2 spatial patches into vectors, which are then projected by a linear layer to match the hidden dimension of the LLM. Conversely, a VAE decoder is employed similarly to reconstruct images from latents. All VAE parameters remain frozen during training.
- Action Input: We employ a simple yet effective linear projection layer as the action encoder, mapping the raw continuous actions directly into the LLM's hidden dimension. Symmetrically, a linear decoder is utilized to project the LLM's output hidden states back to the original action dimensions for execution.

## A.3. Special Tokens

To distinguish different phases of workflow, we add several special tokens to LLM's vocabulary. We list them as follows:

- ⟨subtask_start⟩, ⟨subtask_end⟩
- ⟨plan_start⟩, ⟨plan_end⟩
- ⟨move_start⟩, ⟨move_end⟩
- ⟨action_start⟩, ⟨action_end⟩
- ⟨vision_start⟩, ⟨vision_end⟩

## A.4. Attention Mask

In Figure 3, we have elaborated on the attention mask during fine-tuning phase. The attention masking strategy during pre-training phase is illustrated in Figure 8.

# B. Details on EM-CoT Data Pipeline

## B.1. Action Primitives Extraction

The core extracting procedure is summarized in Algorithm 1.

---

**Algorithm 1** Action Primitives Extraction Algorithm

---

**Input:** Trajectory data $\mathcal{D}$ with gripper states $G^l, G^r$ and end-effector poses $P^l, P^r$; config thresholds $\theta$
**Output:** Frame-wise action labels $\mathcal{L}$ with natural language descriptions
`// Compute instantaneous idle flags` $I^a[t]$ `for arms` $a \in \{l, r\}$`:`
1 **for** $t = 1$ **to** $T - 1$ **do**
2    $I^a[t] \leftarrow \text{IsIdle}(P^a[t], P^a[t-1], G^a[t], G^a[t-1]; \theta_{\text{vel}}, \theta_{\text{dg}})$
3 $I^a[0] \leftarrow \text{True}$
`// Extract action segments via long idle periods:`
4 **for** *arm* $a \in \{l, r\}$ **do**
5    $S^a \leftarrow \text{SegmentActions}(I^a; \theta_{\text{min\_idle}})$

`// Initialize all frames as idle:`
6 **for** $t = 0$ **to** $T - 1$ **do**
7    $\mathcal{L}[t] \leftarrow \{a : [\text{"idle"}, \text{""}] \mid a \in \{l, r\}\}$
`// Label action segments per arm:`
8 **for** *arm* $a \in \{l, r\}$ **do**
9    **for** *each segment* $(s, e) \in S^a$ **do**
      `// Mark gripper actions:`
10       **for** $t = \max(1, s)$ **to** $e$ **do**
11          **if** $\Delta G^a[t] \leq -\theta_{dg}$ **then**
12             $\mathcal{L}[t].a \leftarrow [\text{"grasp"}, \text{""}]$
13          **else if** $\Delta G^a[t] \geq \theta_{dg}$ **then**
14             $\mathcal{L}[t].a \leftarrow [\text{"release"}, \text{""}]$

      `// Label move subsegments:`
15       **for** *each maximal contiguous idle subsegment* $(s', e') \subseteq (s, e)$ **do**
16          $\Delta P = \sum_{t=s'+1}^{e'} (P^a[t] - P^a[t-1])$
         $d \leftarrow \text{GetDirection}(\Delta P; \theta_{\text{dir}})$
         **for** $t = s'$ **to** $e'$ **do**
17             $\mathcal{L}[t].a \leftarrow [\text{"move"}, d]$

`// Convert labels to natural language:`
18 **for** $t = 0$ **to** $T - 1$ **do**
19    **for** *arm* $a \in \{l, r\}$ **do**
20       $\mathcal{L}[t].\text{desc}^a \leftarrow \text{LookupSentence}(\mathcal{L}[t].a)$
21 **return** $\mathcal{L}$

---

**Key Function:**

- `IsIdle()`: Checks if displacement speed $< \theta_{vel}$ and gripper change $< \theta_{dg}$.
- `SegmentActions()`: Finds motion segments separated by idle periods of $\geq \theta_{min\_idle}$ frames.
- `GetDirection()`: Computes dominant motion axes from displacement vector using ratio threshold $\theta_{dir}$.

## B.2. Prompts for EM-CoT

Our annotation pipeline employs a three-stage prompting strategy to transform observations into structured, high-level task descriptions. First, a VLM generates a coherent, temporally-ordered narrative of the entire task. Second, this narrative is

decomposed into a sequence of goal-oriented subtasks. Finally, each video frame is aligned to its corresponding subtask, with the model providing first-person reasoning that links the observed actions to the overall plan. This staged approach ensures temporal coherence and semantic clarity in the final annotations.

---

**Stage 1: Task Narrative Generation**

You are an expert roboticist analyzing a human or robot demonstration. Your task is to write a SINGLE, COHERENT, HIGH-LEVEL NARRATIVE paragraph that STRICTLY follows the TEMPORAL ORDER of the task shown in the video.

- **Overall Goal:** `<instruction>`

- Per-frame low-level actions are provided for context only—DO NOT copy or list them.

- **Low-level Arm Actions:**

    1. Frame_id:0, Left arm action:`<left_description_0>`, Right arm action:`<right_description_0>`
    2. Frame_id:1, Left arm action:`<left_description_1>`, Right arm action:`<right_description_1>`
    3. . . .

**Instructions:**

1. Describe the task step by step in exact chronological order.

2. Explicitly state simultaneous bimanual actions (e.g., "At the same time, the left hand stabilizes... while the right hand unscrews...").

3. Focus on purpose: explain what each action achieves toward the goal.

4. Use specific object names when identifiable.

**Output Rules:** Produce exactly one fluent paragraph (2–4 sentences). Output ONLY the narrative. No markdown, bullets, or extra text.

---

**Stage 2: Subtask Sequence Extraction**

You are an expert in robotic task analysis. Based on the task narrative below, decompose the task into a sequence of HIGH-SEMANTIC, GOAL-ORIENTED SUBTASKS.

- **Task Narrative:** `<narrative>`

**Instructions:**

1. Split the narrative into discrete subtasks in strict chronological order.

2. Represent coordinated bimanual actions as ONE subtask (never split).

3. Express high-level intent (e.g., "Assemble the lid onto the container"), not low-level motions (e.g., "move", "grab").

4. Use imperative, active voice; keep the list short (typically 2–5 subtasks).

**Output Format:** Return ONLY a JSON list of strings. Example: `["Pick up red cup", "Pour water into cup"]`

---

---

**Stage 3: Subtask Alignment and Reasoning**

You are the autonomous onboard controller of a robot. You are currently executing a task. Describe your reasoning in the FIRST PERSON ("I", "me").

- **Overall Goal:** `<instruction>`

- **Planned Subtask Sequence:**

    1. `<subtask_1>`
    2. `<subtask_2>`
    3. . . .

- **Low-level Arm Actions:**

    1. Frame_id:0, Left arm action:`<left_description_0>`, Right arm action:`<right_description_0>`
    2. Frame_id:1, Left arm action:`<left_description_1>`, Right arm action:`<right_description_1>`
    3. . . .

**Instructions:**

1. For each segment, explain your internal decision-making logic from a first-person view.

2. Include: (a) visual observation, (b) goal-driven inference, (c) movement logic.

3. Ensure physical alignment: do not claim a subtask has started if low-level logs show idle.

4. Keep reasoning under 50 words per segment; account for every frame.

**Output Format:** Return ONLY a JSON list of objects with keys: `"subtask"`, `"frame"` (as [start, end]), and `"reasoning"`.

---

## B.3. Visual Subgoal Extraction

---

**Algorithm 2** Visual Subgoal Extraction from Subtask Boundaries

---

**Input:** Frame annotations $\mathcal{A} = \{a_0, a_1, \ldots, a_{T-1}\}$, where $a_t = (\texttt{frame\_id} = t, \texttt{subtask} = s_t)$;
Image sequence $\mathcal{I} = [I_0, I_1, \ldots, I_{T-1}]$
**Output:** Goal image sequence $\mathcal{G} = [G_0, G_1, \ldots, G_{T-1}]$
`// Step 1: Identify goal frame for each subtask`
1 Initialize map $\mathcal{M} \leftarrow \emptyset$ , Sort $\mathcal{A}$ by `frame_id`
2 **for** $t \leftarrow 1$ **to** $T - 1$ **do**
3     **if** $s_t \neq s_{t-1}$ **then**
4        $\mathcal{M}[s_{t-1}] \leftarrow t$
5 $\mathcal{M}[s_{T-1}] \leftarrow T - 1$
`// Step 2: Broadcast goal images to all frames`
6 **for** $t \leftarrow 0$ **to** $T - 1$ **do**
7     $s \leftarrow s_t$ , $t_g \leftarrow \mathcal{M}[s]$ , $G_t \leftarrow I_{t_g}$
8 **return** $\mathcal{G}$

---

# C. Training Implementation

In this section, we provide details on pre-training and fine-tuning of HALO. HALO is pre-trained on 32 Nvidia H100 GPUs for 90k steps, with a sequence length of 40k per rank. Then, we fine-tune HALO on 32 Nvidia H100 GPUs with sequence length of 27k, 110k steps for simulation, 80k steps for real experiment. We illustrate the detailed hyperparameters in Table 4.

*Table 4.* **Training Hyperparameters.** Detailed configurations for the two-stage training process of HALO.

| Configuration | Pre-training | Fine-tuning |
|---|---|---|
| Base Architecture | Qwen2.5-1.5B $\times$ 3 Experts | |
| Total Parameters | $\approx$ 4.5B | |
| Optimizer | AdamW | |
| Learning Rate | $1 \times 10^{-4}$ | $5 \times 10^{-5}$ |
| Learning Rate Schedule | Constant | Constant |
| Weight Decay | 0.0 | 0.0 |
| Per rank Sequence Length | 40k | 27k |
| Warm-up Steps | 2000 | 500 |
| Training Steps | 90k | 110k / 80k |
| Und Resolution | (378, 980) | |
| Gen Resolution | (512, 1024) | |
| Loss Weight | CE:MSE:L1 = 1:2:4 | CE:MSE:L1 = 1:1:1 |

# D. Complete Result on RoboTwin2.0 Benchmark

See Table 5,Table 6 for complete result on RoboTwin 2.0 benchmark.

*Table 5.* Complete Result on RoboTwin (Part 1 of 2)

| Task | $\pi_0$ | | RDT-1B | | Diffusion Policy | | HALO-w/o CoT | | HALO-w/ CoT | |
|---|---|---|---|---|---|---|---|---|---|---|
| | Easy | Hard | Easy | Hard | Easy | Hard | Easy | Hard | Easy | Hard |
| Adjust Bottle | 90 | 56 | 81 | 75 | 97 | 0 | 100 | 11 | 96 | 8 |
| Beat Block Hammer | 43 | 21 | 77 | 37 | 42 | 0 | 73 | 7 | 91 | 4 |
| Blocks Ranking RGB | 19 | 5 | 3 | 0 | 0 | 0 | 71 | 5 | 77 | 8 |
| Blocks Ranking Size | 7 | 1 | 0 | 0 | 1 | 0 | 29 | 2 | 43 | 1 |
| Click Alarmclock | 63 | 11 | 61 | 12 | 61 | 5 | 97 | 6 | 95 | 17 |
| Click Bell | 44 | 3 | 80 | 9 | 54 | 0 | 100 | 26 | 100 | 16 |
| Dump Bin Bigbin | 83 | 24 | 64 | 32 | 49 | 0 | 85 | 39 | 89 | 25 |
| Grab Roller | 96 | 80 | 74 | 43 | 98 | 0 | 99 | 77 | 94 | 46 |
| Handover Block | 45 | 8 | 45 | 14 | 10 | 0 | 78 | 55 | 94 | 31 |
| Handover Mic | 98 | 13 | 90 | 31 | 53 | 0 | 93 | 82 | 99 | 56 |
| Hanging Mug | 11 | 3 | 23 | 16 | 8 | 0 | 22 | 9 | 25 | 5 |
| Lift Pot | 84 | 36 | 72 | 9 | 39 | 0 | 89 | 33 | 95 | 24 |
| Move Can Pot | 58 | 21 | 25 | 12 | 39 | 0 | 87 | 22 | 76 | 14 |
| Move Pillbottle Pad | 21 | 1 | 8 | 0 | 1 | 0 | 66 | 2 | 65 | 13 |
| Move Playingcard Away | 53 | 22 | 43 | 11 | 47 | 0 | 90 | 46 | 87 | 27 |
| Move Stapler Pad | 0 | 2 | 2 | 0 | 1 | 0 | 36 | 15 | 44 | 9 |
| Open Laptop | 85 | 46 | 59 | 32 | 49 | 0 | 76 | 41 | 53 | 21 |
| Open Microwave | 80 | 50 | 37 | 20 | 5 | 0 | 64 | 21 | 20 | 9 |
| Pick Diverse Bottles | 27 | 6 | 2 | 0 | 6 | 0 | 67 | 39 | 76 | 34 |
| Pick Dual Bottles | 57 | 12 | 42 | 13 | 24 | 0 | 84 | 63 | 85 | 52 |
| Place A2B Left | 31 | 1 | 3 | 1 | 2 | 0 | 42 | 15 | 44 | 12 |
| Place A2B Right | 27 | 6 | 1 | 1 | 13 | 0 | 47 | 11 | 55 | 7 |
| Place Bread Basket | 17 | 4 | 10 | 2 | 14 | 0 | 79 | 38 | 75 | 28 |
| Place Bread Skillet | 23 | 1 | 5 | 1 | 11 | 0 | 74 | 33 | 78 | 32 |
| Place Burger Fries | 80 | 4 | 50 | 27 | 72 | 0 | 97 | 13 | 97 | 33 |
| Place Can Basket | 41 | 5 | 19 | 6 | 18 | 0 | 71 | 28 | 38 | 17 |
| Place Cans Plasticbox | 34 | 2 | 6 | 5 | 40 | 0 | 67 | 31 | 83 | 42 |

*Table 6.* Complete Result on RoboTwin (Part 2 of 2)

| Task | $\pi_0$ | | RDT-1B | | Diffusion Policy | | HALO-w/o CoT | | HALO-w/ CoT | |
|---|---|---|---|---|---|---|---|---|---|---|
| | Easy | Hard | Easy | Hard | Easy | Hard | Easy | Hard | Easy | Hard |
| Place Container Plate | 88 | 45 | 78 | 17 | 41 | 0 | 92 | 16 | 92 | 26 |
| Place Dual Shoes | 15 | 0 | 4 | 4 | 8 | 0 | 10 | 1 | 35 | 5 |
| Place Empty Cup | 37 | 11 | 56 | 7 | 37 | 0 | 85 | 6 | 91 | 27 |
| Place Fan | 20 | 10 | 12 | 2 | 3 | 0 | 36 | 0 | 48 | 12 |
| Place Mouse Pad | 7 | 1 | 1 | 0 | 0 | 0 | 36 | 0 | 47 | 8 |
| Place Object Basket | 16 | 2 | 33 | 17 | 15 | 0 | 75 | 4 | 84 | 31 |
| Place Object Scale | 10 | 0 | 1 | 0 | 1 | 0 | 37 | 1 | 38 | 11 |
| Place Object Stand | 36 | 11 | 15 | 5 | 22 | 0 | 85 | 15 | 77 | 45 |
| Place Phone Stand | 35 | 7 | 15 | 6 | 13 | 0 | 75 | 4 | 87 | 27 |
| Place Shoe | 28 | 6 | 35 | 7 | 23 | 0 | 57 | 12 | 75 | 36 |
| Press Stapler | 62 | 29 | 41 | 24 | 6 | 0 | 81 | 29 | 90 | 68 |
| Put Bottles Dustbin | 54 | 13 | 21 | 4 | 22 | 0 | 77 | 26 | 82 | 36 |
| Put Object Cabinet | 68 | 18 | 33 | 18 | 42 | 0 | 44 | 18 | 51 | 19 |
| Rotate QRcode | 68 | 15 | 50 | 5 | 13 | 0 | 54 | 1 | 60 | 11 |
| Scan Object | 18 | 1 | 4 | 1 | 9 | 0 | 62 | 9 | 61 | 21 |
| Shake Bottle Horizontally | 99 | 51 | 84 | 51 | 59 | 18 | 99 | 37 | 99 | 76 |
| Shake Bottle | 97 | 60 | 74 | 45 | 65 | 8 | 99 | 38 | 99 | 75 |
| Stack Blocks Three | 17 | 0 | 2 | 0 | 0 | 0 | 79 | 2 | 66 | 43 |
| Stack Blocks Two | 42 | 1 | 21 | 2 | 7 | 0 | 93 | 12 | 90 | 55 |
| Stack Bowls Three | 66 | 24 | 51 | 17 | 63 | 0 | 66 | 17 | 73 | 30 |
| Stack Bowls Two | 91 | 41 | 76 | 30 | 61 | 0 | 95 | 27 | 94 | 57 |
| Stamp Seal | 3 | 4 | 1 | 0 | 2 | 0 | 30 | 3 | 50 | 19 |
| Turn Switch | 27 | 23 | 35 | 15 | 36 | 1 | 51 | 7 | 54 | 27 |
| **Average** | 46.4 | 16.3 | 34.5 | 13.7 | 28.0 | 0.6 | 70.0 | 21.1 | **72.3** | **27.1** |

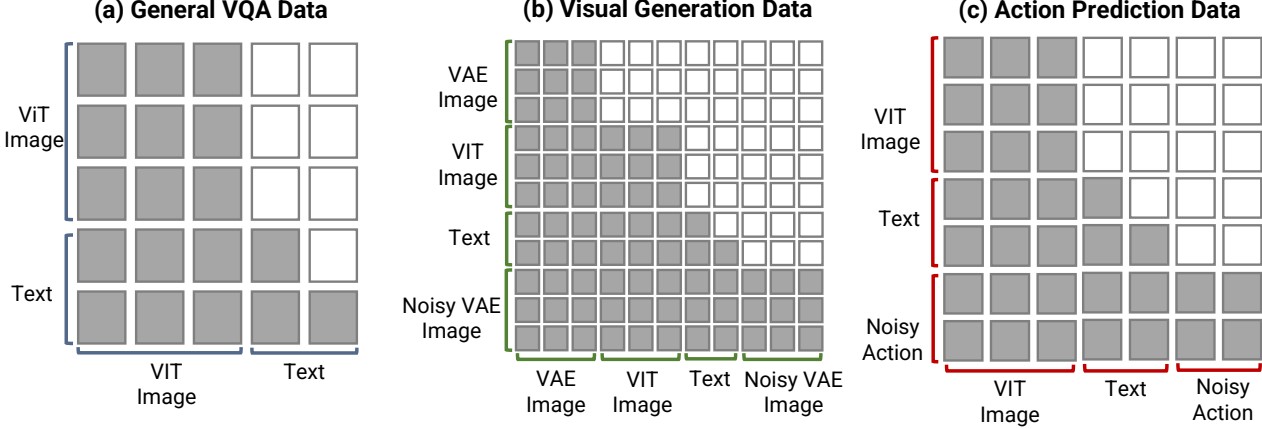

*Figure 8.* Attention map during pre-training phase.

# E. Details on Real-World Experiment

Our real-world experiments are all conducted with COBOT Magic, a robot using the Mobile ALOHA system design (Fu et al., 2024) and manufactured by AgileX Robotics. Notably, we exclusively utilize the robot's head camera as the sole source of visual input, requiring the model to reason and act based on an egocentric perspective. To evaluate policies, we design a set of tasks that span various dimensions, including:

- **Tool-mediated Manipulation**: Gently pick up the broom and sweep the buttons on the board into the dustpan. ("Sweep the buttons")
- **Semantic Visual Grounding**: Place the grey cup into the pink cup with left gripper, and the green cup into the yellow cup with right gripper. ("Bimanual cup nesting")
- **Inter-arm Collaboration**: Handover the screwdriver from the left to the right. ("Handover the screwdriver")
- **Multi-step Manipulation**: Open the top drawer, put the lemon on the plate into the top drawer, close the drawer. ("Put lemon into drawer")

We collect 80 trajectories for each task, and process them through our automated pipeline to yield the EM-CoT data. We then aggregate the data across all four tasks into a multi-task corpus for finetuning.

In order to evaluate generalization, we design settings that span various axes of unseen scenarios, including visual distraction, lighting variation, background variation, and novel object. The generalization settings are formulated as follows.

- **Visual Distraction**: Randomly placing irrelevant objects such as Coke bottle, cups onto the table.
- **Lighting Variation**: Varying the illumination levels of a desk lamp to introduce visual interference.
- **Background Variation**: Changing the color of the tablecloth.
- **Novel Object**: For the task of sweeping the buttons, we change the broom to a sponge during evaluation.

