# OpenReview forum: "HALO: A Unified Vision-Language-Action Model for Embodied Multimodal Chain-of-Thought Reasoning"
_ICML.cc/2026/Conference — ICML 2026 regular_

### Official Review · Reviewer_S8VU · 2026-03-03

**Soundness:** 2
**Presentation:** 2
**Significance:** 3
**Originality:** 3
**Overall Recommendation:** 3
**Confidence:** 3

**Summary:**

The authors propose the HALO model, which innovatively introduces Embodied Multimodal Chain-of-Thought (EM-CoT). This approach simulates human cognition and execution through a sequential process of “text reasoning → visual subgoal prediction → action prediction.”
Architecturally, HALO adopts a Mixture of Transformers (MoT) design, decoupling semantic understanding, visual generation, and action prediction into three specialized expert modules. In addition, the authors develop an automated EM-CoT data synthesis pipeline and a two-stage training strategy.
Experimental results demonstrate that HALO achieves significant performance improvements on both the RoboTwin 2.0 simulation platform and real-world robots (Mobile ALOHA), for example outperforming the $\pi_0$ baseline by 34.1% in simulation.

**Compliance With Llm Reviewing Policy:**

Affirmed.

**Final Justification:**

While I appreciate the paper's innovative attempt in the direction of embodied multimodal CoT, after a comprehensive evaluation, I believe the work has not yet reached the full maturity required for acceptance. Therefore, I have decided to maintain my score of 3.

**Key Questions For Authors:**

1. Please provide the specific inference frequency (Hz) of HALO when deployed in the real world. Compared to the $\pi_0$ model that directly outputs actions, how much additional time overhead is introduced by generating EM-CoT (especially the high-dimensional visual sub-goals)? In practical testing, does this latency cause noticeable stuttering or delays in robot motion?

2. If the visual generation expert predicts a seemingly reasonable but physically implausible (or infeasible in the current environment) sub-goal image, how does the action prediction expert respond? Does the model have any mechanism to correct such erroneous “imaginations”?

3. Please explicitly report in the appendix the parameter scales of baseline models such as $\pi_0$ and RDT, as well as the pretraining data they use. If there are discrepancies, please explain why the comparison remains fair without aligning computational resources.

4. Can the rule-based extractor in Algorithm 1 handle manipulation tasks in continuous trajectories where there is no clearly defined “Idle” state?

**Limitations:**

The code and real-world robot demo are not provided.

The current visual foresight mechanism generates only a single terminal subgoal image (Terminal Frame) for each subtask, rather than modeling continuous video-level forward dynamics. In contact-intensive tasks, changes in intermediate states (e.g., the deformation process of objects) are often more critical than the final state itself. A single image may therefore be insufficient to provide adequate fine-grained guidance.

**Strengths And Weaknesses:**

**Strengths**

Adopting a Mixture of Transformers (MoT) architecture is a particularly notable design choice. It not only avoids mutual interference between generative tasks across different modalities (e.g., the degradation of a VLM’s textual reasoning ability due to visual token prediction), but also ensures smooth cross-modal information interaction.

The paper proposes an EM-CoT data synthesis pipeline that requires no manual intervention. By leveraging an off-the-shelf VLM (e.g., Qwen3-VL) and heuristic rules, low-level robotic trajectories are transformed into structured data pairs in the form of “high-level task → visual subgoal → low-level action.”

Extensive experiments are conducted in both simulation (RoboTwin 2.0, comprising 50 tasks) and the real world (4 long-horizon tasks under various disturbance settings). The ablation studies clearly demonstrate the necessity of multimodal chain-of-thought modeling (text-only vs. vision-only vs. combined) as well as different combinations of pretraining data.

Both qualitative and quantitative analyses show that, when facing unseen backgrounds, lighting conditions, and distractors, the model significantly outperforms traditional reactive baselines, largely due to the guidance provided by its explicit reasoning chain.

**Weaknesses**

EM-CoT requires the model to autoregressively generate textual reasoning before outputting the Action Chunk, and then to decode visual sub-goal images via diffusion or a VAE. In real-world deployment, this “think–imagine–act” paradigm may introduce substantial latency, limiting its applicability in tasks that demand high-frequency reactions or dynamic obstacle avoidance.

In Algorithm 1, the extraction of action primitives heavily relies on manually defined thresholds (e.g., velocity thresholds, gripper open/close thresholds, and minimum idle time). While this rule-based segmentation approach is effective for discrete pick-and-place tasks, its suitability for fluid manipulation, non-prehensile contact (e.g., pushing or wiping), or highly dynamic tasks remains questionable.

HALO contains approximately 4.5 billion parameters and is pretrained on large-scale heterogeneous datasets (LLaVA, SSv2, OXE). If the baselines are smaller in scale, the observed performance gap may partially stem from differences in model capacity and data scale, rather than from the intrinsic advantages of the EM-CoT mechanism itself.

---

> ### Author Rebuttal · Authors · 2026-03-31
>
> ### W1 & Q1: Does EM-CoT introduce substantial inference latency?
>
> In practice, HALO operates at **10 Hz** with a chunk size of 50. With EM-CoT enabled, the total inference time is **3.1s** (1.3s preprocessing, 0.3s text generation, 0.9s image generation, and 0.6s action generation), compared to **0.6s** for action generation alone without EM-CoT. Importantly, we did **not observe perceptible stuttering or delay** in real-world deployment.
>
> This is enabled by two factors. First, HALO uses **asynchronous execution**: each 50-step action chunk takes about **5s** to execute, and EM-CoT reasoning for the next chunk is computed in parallel during this window. Once the next chunk is ready, execution switches seamlessly. Second, we apply **system-level inference optimizations** so that the full EM-CoT pipeline fits within the execution window.
>
> We will clarify this deployment setup and latency analysis in the revision.
>
> ---
>
> ### W2 & Q4: Can the rule-based extractor in Alg.1 handle trajectories without a clear “Idle” state?
>
> The use of the **Idle** state is mainly motivated by the **RoboTwin** dataset, where trajectories can be naturally segmented by Idle periods; within each segment, the action primitive is assigned by majority voting.
>
> However, this design is **not essential**. In real-world experiments without explicit Idle segmentation, we instead use a **sliding window** (window size 4) to compute action primitives and assign the resulting primitive to the first frame in the window.
>
> We will update Algorithm 1 in the revision to reflect this more general formulation.
>
> ---
>
> ### W3 & Q3: Does HALO’s performance gain stem from its design, or from differences in model capacity and data scale?
>
> HALO’s gains come from its **pretraining strategy** and **EM-CoT mechanism**, rather than simply scaling data or compute. In fact, both **π₀** and **RDT** are pretrained with substantially larger datasets and more compute than HALO.
>
> We summarize the pretraining statistics below:
>
> | Metric | $\pi_0$ | RDT | HALO |
> |---|---:|---:|---:|
> | **\# Param** | 3.3B | 1.2B | 4.5B |
> | Data Scale | 797K timesteps, $\sim$100 TB | 1M+ trajectories, 21 TB | 2 TB |
> | Data Details | OXE, Pi dataset | 46 Open Robotic Datasets (including OXE) | OXE, LLaVA-NeXT, SSv2 |
> | Pre-train Compute | $\sim$256 H100 for several weeks | 48 H100 for 1 month | 32 H100 for 1 week |
>
> Despite using significantly fewer resources, HALO achieves state-of-the-art performance. We attribute this to two main design choices:
> (1) the **unified MoT architecture**, which preserves modality-specific features while enabling cross-modal interaction through shared self-attention, and
> (2) **EM-CoT fine-tuning**, which unlocks the model’s reasoning capability in a think–imagine–act manner.
>
> We will make this comparison and attribution more explicit in the revision.
>
> ---
>
> ### Q2: How robust is the action expert to implausible visual sub-goals? Are there error-correction mechanisms?
>
> Implausible intermediate reasoning does not necessarily lead to task failure. As shown in **Table 2 (Panel B)**, EM-CoT improves performance by **+5.2%** overall.
>
> We further analyzed **100 trials** on the RoboTwin *Move Pillbottle Pad* task. Among the **83** trajectories with plausible EM-CoT, HALO achieves **80.7%** success. Among the remaining **17** trajectories with implausible visual sub-goals, HALO still achieves **52.9%** success. This demonstrates that the action expert is reasonably robust to imperfect intermediate predictions.
>
> In addition, HALO mitigates error accumulation by **regenerating EM-CoT at each step** based on the latest observation, allowing the policy to correct earlier mistakes online.
>
> ---
>
> ### L1: The code and real-world robot demo are not provided.
>
> We will release the **code, checkpoints, and real-world robot demo videos** soon.
>
> ---
>
> ### L2: Video-level foresight for contact-intensive tasks.
>
> Thank you for this insightful suggestion. We agree that **video-level foresight** could be beneficial for contact-rich tasks. However, for most tasks in our current setting, predicting a **subgoal image** is more efficient and practical, while video prediction introduces higher computational cost and additional latency.
>
> Since this work focuses on **Embodied CoT**, we adopt image-based foresight as a better trade-off between reasoning capability and deployment efficiency. We will clarify this limitation and include **video-level foresight** as an important direction for future work.

---

> > ### Author Rebuttal · Reviewer_S8VU · 2026-04-01
> >
> > Thank you to the authors for their detailed and thoughtful responses. I will be maintaining my original score.

---

> > > ### Author Response · Authors · 2026-04-03
> > >
> > > Thank you for acknowledging our rebuttal. We are glad that our responses addressed your concerns.
> > > Your final score is very important to this work. If you still have any remaining concerns, please let us know, and we will do our very best to address them. We sincerely hope you could consider raising your score.

---

### Official Review · Reviewer_WLLY · 2026-03-09

**Soundness:** 3
**Presentation:** 3
**Significance:** 3
**Originality:** 2
**Overall Recommendation:** 3
**Confidence:** 4

**Summary:**

This paper introduces HALO, a unified Chain-of-Thought (CoT) Vision-Language-Action (VLA) model. The core architecture leverages a Mixture-of-Transformers framework to co-train three distinct tasks: Language-CoT reasoning, sub-goal generation (Visual-CoT), and action prediction. HALO achieves state-of-the-art (SOTA) performance on both the Robotwin simulation and real-world robotic benchmarks. Notably, the authors demonstrate that enabling the CoT mechanism leads to a measurable improvement in policy performance.

**Compliance With Llm Reviewing Policy:**

Affirmed.

**Final Justification:**

The VLA model proposed in this paper is fine-tuned on a 1.5B LLM backbone. In the field of embodied AI, a model of this scale is typically expected to have a certain level of generalizability, especially since the CoT method was specifically designed to address OOD challenges. Currently, VLA models are still in an exploratory phase, and it remains unclear how CoT should be implemented or how effective it truly is. I believe the research community is not simply looking for "another VLA model," but rather for work that offers a deep analysis of the CoT mechanism. Consequently, I hope the authors would provide visualizations of their model’s CoT performance (including both generated subgoal images and language instructions) during real-robot OOD tasks. Unfortunately, these were not provided. For these reasons, I will maintain my rating of weak reject.

**Key Questions For Authors:**

1. Could the authors provide a more granular analysis of HALO’s internal reasoning, as suggested in the weaknesses?

2. Could you provide qualitative examples of the CoT reasoning and sub-goal generation when the model is faced with a task or environment not seen during training? Does the reasoning remain physically grounded and logical in these scenarios?

**Limitations:**

yes

**Strengths And Weaknesses:**

### Strengths

1. The paper is easy to follow, and the figures are high-quality and visually informative, aiding in the understanding of the complex system.

2. The proposed architecture is sophisticated and involves significant engineering effort. If the authors release the model and codebase, it would constitute a substantial contribution to the Embodied AI community.

3. The experimental findings confirm that incorporating CoT reasoning enhances the base policy's performance, providing further evidence for the importance and effectiveness of reasoning-based paradigms in robotics.

---

Weaknesses

1. Currently, the paper reads primarily as a large-scale engineering implementation that yields strong results, but it offers limited scientific insight. The community would benefit from understanding the "why" behind these results. For instance: (a) How does the Language-CoT analyze an out-of-distribution (OOD) task? (b) What specific types of sub-goals does the image model generate during execution? (c) What is the individual contribution of each component? (d) Specifically, what is the performance gain when using only Language-CoT versus only Visual-CoT? How do these two modalities interact to influence the final policy? Without a clear ablation study or qualitative analysis of these mechanisms, it is difficult for practitioners to justify adopting the complex Mixture-of-Transformers architecture over more established, validated frameworks like the $\pi$-series.

2. Minor Presentation Error: In Figure 7, the two sub-figures appear to be swapped. Please verify and correct the labeling/placement.

---

> ### Author Rebuttal · Authors · 2026-03-31
>
> ### W1: Analysis and Ablation of HALO's Design Choices
> We respectfully note that many requested analyses already exist in our paper: **Panel A & B** (**Sec. 4.3**) ablate architectural components and individual CoT contributions, **Sec. 4.4** visualizes OOD reasoning, and **Sec. 3.2** discusses cross-modal interaction. We further elaborate on each point below.
>
> (a) Language-CoT for OOD tasks.
> Referring to the Qualitative Results in Sec. 4.4, Language-CoT actually exhibits robust OOD generalization. To support this, we visualize our results on the RoboTwin hard settings, which feature heavy randomization unseen during training. Results show that HALO is able to precisely generate textual reasoning in these OOD settings, especially tasks with well-defined steps such as Blocks ranking RGB. This is a natural outcome, since textual reasoning operates in semantic space, inherently decoupled from visual perturbations like changed textures or background.
>
> (b) Specific type of visual sub-goals.
> Referring to **Sec. 3.3 and Appendix B.3**, we define visual subgoals as the terminal keyframes of each subtask, representing the target world state for subtask completion. In contrast to fixed-interval sampling, this design achieves close semantic alignment between visual and textual modalities, mirroring the human cognitive process of 'imagining key events'. We empirically verified that keyframe subgoals outperform fixed-interval alternatives, as the latter often capture mid-transition moments with ambiguous goal semantics rather than clear target states.
>
> (c)(d) Individual component contributions.
> The fundamental logic of EM-CoT is the think–imagine–act paradigm, in which each modality serves as an integral component of a cohesive reasoning chain. Table 2 provides a systematic ablation from two complementary perspectives. Panel A validates the architecture: progressively removing the pretraining stage of each expert (Visual → Textual → Action) leads to consistent degradation (75.3 → 58.2 → 42.9 → 32.4), confirming that the pretraining recipe for every expert is essential. Panel B validates the EM-CoT reasoning chain itself: removing textual planning (w/o T) drops Hard from 26.4 to 18.3, removing visual foresight (w/o V) drops Easy from 80.5 to 76.1, and removing both reduces performance to that of the base model (80.5 → 75.3 / 26.4 → 21.2). Together, the two panels confirm that the architecture and the reasoning mechanism are each indispensable.
>
> (e) Interaction of language and vision modalities.
> This interaction is discussed in Sec 3.2, where we elaborate on how the shared self-attention and carefully-tailored attention mask facilitate cross-modal alignment.
>
> (f) It is difficult for practitioners to justify adopting the MoT architecture over frameworks like the $\pi$-series.
> Mixture-of-Transformers (MoT) refers to a design where each transformer experts maintain independent weight while sharing the self-attention mechanism. Therefore, the **$\pi$-series** is also a MoT model with distinct VLM expert and action expert. The uniqueness of HALO is that the dedicated visual expert endows the policy with world modeling capability. Removing it alone causes a significant drop (**75.3→58.2 Easy**, **21.2→10.5 Hard**), confirming that visual grounding is critical beyond architectural design. Building on this visual dynamics modeling, EM-CoT further enables a "think-imagine-act" reasoning chain, where textual planning and visual foresight jointly ground every action output.
>
> ### W2: Minor Presentation Error
> We thank the reviewer for catching this oversight. We will correct Fig. 7 in our revised manuscript.
>
> ### Q1: More granular analysis of HALO’s internal reasoning.
> We highlight three key takeaways regarding HALO's internal 'think-imagine-act' workflow. Hope these insights address your concern.
> * Textual reasoning for OOD: Language CoT reasons semantically, making it naturally invariant to visual shifts. Results show that removing it causes the largest Hard-task drop (26.4%→18.3%).
> * Visual foresight for spatial grounding: Subgoal images translate abstract plans to concrete spatial targets, complementing **what to do with where**. Removing them mainly hurts Easy tasks (80.5%→76.1%), which often requires fine-grained actions.
> * Layered conditioning: Text narrows task intent, subgoals narrow spatial targets, forming coarse-to-fine guidance that especially benefits long-horizon tasks (e.g., Blocks Ranking Size **+21%**).
>
> ### Q2: Qualitative examples of EM-CoT in OOD scenarios. Does the reasoning remain physically grounded and logical?
> Fig. 5 already provides examples under both clean and OOD (unseen domain-randomized) settings. Despite heavy visual perturbations, textual reasoning correctly decomposes tasks and describes scene states, while generated subgoal images, though slightly less sharp, still depict the correct target configurations. We will include additional OOD examples in our revision.

---

> > ### Author Rebuttal · Reviewer_WLLY · 2026-04-03
> >
> > I would like to thank the authors for their detailed response. I believe there may be some misunderstandings regarding my original comments. First, I appreciate the extensive feedback on W1; however, my intention was merely to suggest more in-depth analysis rather than expecting a complete set of new experiments within the limited rebuttal window.
> >
> > Furthermore, while I am fully aware of the ablation studies and comparisons already presented in the manuscript, my concern regarding OOD generalization does not refer to the clean vs. randomized settings in RoboTwin. I am referring to the real-robot experiments. For instance, in your task like "placing an object into a drawer," I am interested in how the model performs if the object is replaced by a random, unseen item with significant morphological differences. Specifically:
> >
> > 1. Can the Language-CoT still accurately describe the novel object and decompose the task correctly?
> > 2. Can the Image-CoT generate a plausible and correct subgoal image?
> >
> > This type of stress test better reflects the challenging scenarios VLA models encounter in real-world applications. Such evidence is crucial to demonstrate the necessity of CoT and to justify the additional computational overhead during inference. These qualitative visualizations would provide deeper insights into the underlying mechanism of CoT and could potentially be more persuasive than mere changes in success rates.

---

> > > ### Author Response · Authors · 2026-04-07
> > >
> > > We thank the reviewer for the insightful follow-up. We provide a more in-depth analysis of EM-CoT behavior under real-robot novel object scenarios below.
> > >
> > > For both Q1 and Q2, our experiments confirm that EM-CoT can in most cases accurately describe novel objects, decompose the task correctly, and generate plausible subgoal images. We support this with two pieces of evidence. First, our existing generalization experiment in Fig. 7(b) already evaluates object substitution in the sweeping task (broom to sponge), where HALO with EM-CoT achieves 58% success rate compared to 54% for HALO-w/o EM-CoT, with Language-CoT correctly identifying the novel object and Image-CoT generating spatially correct subgoal images. Second, we conduct a new real-robot experiment replacing the lemon with an apple in the drawer task. The results show that Language-CoT still correctly decomposes the task since the subtask structure is object-agnostic and remains intact, and Image-CoT generates a subgoal image where the spatial configuration is correct despite some degradation in visual sharpness.
> > >
> > > That said, we do observe occasional failures in object identification in some steps. Crucially however, EM-CoT exhibits a self-correction mechanism: since each step's reasoning is conditioned on the latest observation, a mis-identification in one step can be naturally corrected in subsequent steps as the model receives updated visual feedback. This iterative grounding between perception and reasoning is a key advantage of the chain-of-thought formulation over reactive policies, which have no such recovery mechanism once an error occurs.
> > >
> > > We commit to including these qualitative visualizations in the revision, shown side-by-side with the in-distribution case, which we hope will provide the mechanistic insight the reviewer is looking for beyond success rates alone.
> > >
> > > ---
> > >
> > > ### Update: Anonymous Supplementary Visualizations Now Available
> > >
> > > Following our previous commitment, we have now prepared an **anonymous supplementary page** containing the qualitative real-robot EM-CoT visualizations, made available here for the reviewer's convenience during the final assessment:
> > >
> > >  **https://anonymous.4open.science/r/HALO_Realexp_vis-BA40/halo_real_visualization.pdf**
> > >
> > > The page contains side-by-side visualizations of both **textual reasoning traces** and **generated subgoal images** on the real robot, under both **in-distribution** and **generalization** conditions, directly corresponding to the three mechanistic aspects discussed above:
> > >
> > > 1. **Language-CoT decomposition under novel objects** (orange → green apple): the subtask structure remains object-agnostic and stays intact under appearance shifts;
> > > 2. **Image-CoT spatial grounding** (marker → screwdriver): the generated subgoals still capture the correct target configuration and anchor downstream action prediction, despite some degradation in visual sharpness;
> > > 3. **Self-correction behavior** (grey clothes → red clothes): occasional mis-identifications in one step are naturally recovered in subsequent steps as the model iteratively grounds on the latest observation — an advantage reactive policies fundamentally lack.
> > >
> > > We hope these targeted visualizations help address your remaining concern about how EM-CoT behaves in real-robot novel-object scenarios. We would be grateful if you could take this additional evidence into account in your final assessment.

---

### Official Review · Reviewer_6zLc · 2026-03-12

**Soundness:** 3
**Presentation:** 3
**Significance:** 3
**Originality:** 3
**Overall Recommendation:** 5
**Confidence:** 4

**Summary:**

This paper addresses the limitations of existing vision-language-action models in long-horizon and out-of-distribution robotic manipulation, arguing that direct action prediction lacks explicit mechanisms for reasoning and anticipating future world states. It proposes HALO, a unified model for embodied multimodal chain-of-thought reasoning that performs sequential textual planning, visual subgoal prediction, and action generation using a Mixture-of-Transformers architecture with specialized experts and shared self-attention. The model is trained with a two-stage recipe combining broad multimodal pretraining and fine-tuning on automatically synthesized EM-CoT data derived from robot trajectories. Evaluation on the 50-task RoboTwin 2.0 benchmark and on four real-world manipulation tasks shows reported improvements over baselines including π0, RDT-1B, and Diffusion Policy, with ablations suggesting that both textual reasoning and visual foresight contribute to performance and generalization.

**Compliance With Llm Reviewing Policy:**

Affirmed.

**Final Justification:**

The authors have successfully addressed my concerns. I will raise my rating from Weak accept to Accept.

**Key Questions For Authors:**

1. Could you conduct an ablation study that exchanges the order of text and image sequences in the CoT process? I am particularly interested in understanding how the ordering of multimodal CoT components affects model performance.
2. Could you provide a comprehensive analysis of the quality of EM-CoT data?

**Limitations:**

yes

**Strengths And Weaknesses:**

**Strength**

1. The paper introduces a clearly structured unified architecture that decouples semantic reasoning, visual foresight, and action prediction into specialized transformer experts while enabling interaction through shared self-attention. This design is technically well specified, including explicit routing tokens and masking strategies, and is motivated as a way to preserve modality-appropriate generation mechanisms such as autoregressive text generation and diffusion-style prediction for images and actions.
2. The empirical evaluation is extensive across both simulation and real-world settings. In simulation, HALO is evaluated on 50 RoboTwin 2.0 tasks under both Easy and domain-randomized Hard settings, where the paper reports average success rates of 80.5% and 26.4%, respectively, compared with 46.4% and 16.3% for π0; the inclusion of HALO-w/o EM-CoT also helps isolate the effect of the reasoning formulation.
3. The paper provides focused ablation studies that give concrete support to the proposed training recipe and EM-CoT design. Table 2 shows that removing visual generation pretraining substantially reduces performance (from 75.3/21.2 to 58.2/10.5 on Easy/Hard), and that removing either textual reasoning or visual subgoals from EM-CoT also lowers success rates relative to full HALO, supporting the claim that both modalities contribute to the final gains.

**Weakness**

1. The paper would benefit from a more rigorous analysis of how multimodal CoT affects reasoning efficiency, since the current presentation does not clearly quantify whether it improves decision quality, computational cost, or inference latency.
2. Although the main focus is on dual arm VLA models, it would be valuable to include evaluations on more general single arm benchmarks by disabling the left arm during execution and comparing against existing pure vision CoT methods [1-3]. Such experiments would help clarify the specific contribution of multimodal joint CoT and strengthen the empirical scope of the paper.
3. The paper lacks ablation studies to validate the effectiveness of the multi-stage training recipe. In particular, it would be helpful to report the performance of a single-stage full-task training setting as a baseline. Such a comparison would provide clearer evidence for the advantage of the multi-stage training strategy.

[1] Qingqing Zhao, Yao Lu, Moo Jin Kim, Zipeng Fu, Zhuoyang Zhang, Yecheng Wu, Zhaoshuo Li, Qianli Ma, Song Han, Chelsea Finn, Ankur Handa, Ming-Yu Liu, Donglai Xiang, Gordon Wetzstein, Tsung-Yi Lin. CoT-VLA: Visual Chain-of-Thought Reasoning for Vision-Language-Action Models. In Proc. of CVPR 2025.

[2] Qi Lv, Weijie Kong, Hao Li, Jia Zeng, Zherui Qiu, Delin Qu, Haoming Song, Qizhi Chen, Xiang Deng, Jiangmiao Pang. F1: A Vision-Language-Action Model Bridging Understanding and Generation to Actions. In Proc. of NeurIPS 2025.

[3] Yuqi Wang, Xinghang Li, Wenxuan Wang, Junbo Zhang, Yingyan Li, Yuntao Chen, Xinlong Wang, Zhaoxiang Zhang. Unified Vision-Language-Action Model. In Proc. of ICLR 2026.

---

> ### Author Rebuttal · Authors · 2026-03-31
>
> ### W1: Quantitative efficiency analysis of EM-CoT.
>
> We evaluate EM-CoT's efficiency across three dimensions:
>
> - **EM-CoT substantially improves decision quality.** EM-CoT brings +5.2% Easy / +5.2% Hard over HALO-w/o EM-CoT (Table 2 Panel B), with especially large gains on long-horizon tasks that require multi-step planning (e.g., Stack Blocks Three +6%, Stamp Seal +11%).
> - **Training overhead remains manageable.** HALO requires ~50× less pre-training data and ~8× fewer GPUs than $\pi_0$. EM-CoT fine-tuning adds 2 extra days (from 2 to 4 days on 32 H100s), a manageable overhead given substantial performance gains.
> - **Inference latency is negligible.** Through engineering optimizations, reasoning is compressed to 3.1s and fully absorbed via asynchronous execution during the action chunk window, maintaining 10 Hz real-time control with no perceptible delay.
>
> Overall, EM-CoT delivers significant decision quality gains at modest training cost and without perceivable deployment overhead.
>
> ---
> ### W2: Evaluation on single-arm benchmarks and comparison with vision-only CoT methods.
>
> We appreciate the reviewer's constructive suggestion. We note that the RoboTwin benchmark already contains **28 single-arm tasks** (e.g., Adjust Bottle, Click Bell, Shake Bottle, etc.), where only one arm is active. On these tasks, HALO achieves **79.3% / 24.8%** on clean / hard settings, outperforming HALO-w/o EM-CoT by **+4.5% / +5.7%** respectively. This confirms that EM-CoT's benefits are not limited to bimanual coordination but extend equally to single-arm tasks.
>
> Referring to our ablation in **Table 2, Panel B**, the w/o T variant (visual subgoal only, analogous to pure vision CoT) achieves 77.8% / 18.3%, while the full EM-CoT reaches **80.5% / 26.4%**, confirming that multimodal joint CoT provides clear gains over vision-only reasoning. Due to the limited rebuttal period, we are unable to collect such EM-CoT annotations and retrain on new single-arm benchmarks, but we will include such comparisons with [1-3] in our future revision.
>
> ---
> ### W3: Single-stage vs. Multi-stage training ablation.
>
> We trained HALO directly on EM-CoT data **without pre-training** and evaluated it on a randomly selected 10-task subset of RoboTwin (due to the considerable time required for full evaluation). Performance on the easy setting drops sharply to **11%**, while success on the hard setting collapses to near **0%**. Upon carefully examining the generated CoT, we find that while textual reasoning is acquired relatively easily, the visual expert without pre-training produces **severely distorted and semantically meaningless subgoal images**, which irreversibly misguide action prediction. This validates the necessity of our two-stage training paradigm: the full potential of EM-CoT reasoning can only be unlocked when built upon a solid foundation established through **extensive pre-training**.
>
> ---
> ### Q1: Text-image ordering in EM-CoT.
>
> We have **changed the order of textual reasoning and visual subgoal** within EM-CoT, and re-run our training. Testing on the 10-task subset of RoboTwin, performance drops to **54%** on easy setting and **9%** on hard setting. We attribute this to the disruption of the **natural coarse-to-fine reasoning flow**: textual reasoning first narrows the task-level intent (e.g., "pick up the red block"), which then guides the visual expert to generate a spatially grounded subgoal consistent with that plan. Reversing this order forces the visual expert to generate subgoals without semantic guidance, producing less accurate images that in turn corrupt the downstream textual reasoning and action prediction. This confirms that the **text→image→action** ordering is not arbitrary but reflects a principled information hierarchy essential to EM-CoT's effectiveness.
>
> ---
> ### Q2: Quality assessment of EM-CoT data.
>
> We assess EM-CoT data quality from three complementary angles:
>
> - **Textual annotation quality.** To verify extraction correctness, we visualized action primitives via video overlays and conducted human evaluation on 100 sampled trajectories, achieving 92% subtask boundary accuracy and 89% reasoning faithfulness.
> - **Visual subgoal quality.** Since subgoals are defined as ground-truth terminal frames of each subtask (Algorithm 2) rather than model-generated, their quality is directly determined by subtask boundary accuracy, which was validated above.
> - **Downstream validation.** EM-CoT consistently improves across all 50 tasks (+5.2% Easy / +5.2% Hard), and qualitative results (**Figure 5**) show faithful reasoning reproduction even under OOD scenarios — strong indirect evidence of high data quality.

---

> > ### Author Rebuttal · Reviewer_6zLc · 2026-04-03
> >
> > The authors have addressed the concerns. I am satisfied with their responses and will maintain my original rating.

---

> > > ### Author Response · Authors · 2026-04-03
> > >
> > > Thank you for your thoughtful review, constructive feedback, and positive rating. We are glad that our responses have addressed your concerns. Please feel free to reach out if you have any further questions.

---

### Official Review · Reviewer_DxBt · 2026-03-13

**Soundness:** 4
**Presentation:** 3
**Significance:** 3
**Originality:** 3
**Overall Recommendation:** 5
**Confidence:** 3

**Summary:**

This paper proposes HALO, a unified VLA model that enables embodied multimodal chain-of-thought (EM-CoT) reasoning. The model decomposes decision-making into three stages: textual reasoning, visual subgoal prediction, and action generation. A Mixture-of-Transformers (MoT) architecture is introduced to decouple multimodal understanding, visual generation, and action prediction into specialized experts with shared attention. The authors further design an automated EM-CoT data synthesis pipeline and a two-stage training strategy. Experiments on RoboTwin 2.0 and real-world robotic tasks show substantial improvements over strong baselines, particularly in long-horizon and domain-randomized settings.

**Compliance With Llm Reviewing Policy:**

Affirmed.

**Final Justification:**

I maintain my original score and recommendation for acceptance.

**Key Questions For Authors:**

It would strengthen the paper to more clearly position EM-CoT with respect to foundational reasoning-and-acting frameworks (e.g., ReAct, SayCan, DWIM) and recent decoupled or tool-augmented VLA systems, clarifying the precise theoretical distinction and novelty of the proposed formulation.

**Limitations:**

yes

**Strengths And Weaknesses:**

Strengths
- Well-motivated unified framework integrating reasoning, foresight, and control.
- MoT architecture effectively decouples heterogeneous generation processes while enabling cross-modal interaction.
- Clear training recipe with versatile pretraining + EM-CoT fine-tuning.
- Strong empirical results with significant gains over π0 and other baselines.
- Real-world validation demonstrating robustness and generalization.

Weakness
- Broader conceptual positioning could be strengthened. While the paper focuses on embodied VLA models, it would benefit from a clearer discussion of related reasoning-and-acting frameworks beyond the immediate embodied literature. In particular, paradigms such as ReAct (ICLR 2023), SayCan (PMLR 2023), and more recent system-driven or decomposition-based approaches (e.g., DWIM, ICCV 2025) also explore structured reasoning interleaved with action execution. Although these works are not strictly unified VLA models, discussing their conceptual similarities and differences would help better contextualize the theoretical contribution of EM-CoT.

- System complexity vs. benefit trade-off.
The MoT architecture and EM-CoT pipeline introduce additional architectural and training complexity. A clearer analysis of efficiency, scalability, or inference overhead relative to simpler VLA baselines would further strengthen the empirical narrative.

---

> ### Author Rebuttal · Authors · 2026-03-31
>
> ### W1 & Q1: Broader positioning of EM-CoT within foundational reasoning-and-acting frameworks.
>
> We sincerely thank the reviewer for this constructive insight. In our view, existing reasoning-and-acting frameworks share a common principle: leveraging **explicit reasoning** to distill **foundation model priors** into tractable subgoals, thereby **compressing the vast action space** for downstream execution. In fact, the three works the reviewer mentioned each instantiate this principle from a distinct angle:
>
> **SayCan**[1] grounds LLM planning in **robotic tasks** via affordance-based skill ranking, but remains **open-loop**. **ReAct**[2] brings reasoning-acting interleaving to **language agents** with **closed-loop feedback**, but assumes reliable tool outputs. **DWIM**[3] targets compositional **visual reasoning** by detecting tool-output discrepancies to recover more training data and selectively learning from effective actions, turning a frozen LLM into a **tool-aware agent**.
>
> Beyond these reasoning-and-acting frameworks, some decoupled robotic systems also pursue structured reasoning: **Code as Policies**[4] shows that LLMs can directly compose robot primitives via code generation, bypassing the need for task-specific planners; **VoxPoser**[5] demonstrates that LLM+VLM can jointly synthesize 3D affordance maps, grounding language reasoning in spatial geometry for motion planning. However, both rely on modular pipelines where **planning and execution remain separate**.
>
> **HALO** inherits the core insight from these prior works, but with two key distinctions: (1) it unifies textual reasoning and visual foresight within **a single end-to-end model** rather than modular pipelines, and (2) its reasoning directly produces **continuous motor commands**, closing the full loop from deliberation to physical execution. We believe extending HALO with *closed-loop reasoning-acting interleaving* (as in ReAct/DWIM) in embodied settings is a promising future direction. We will incorporate the aforementioned works into our revised related work section to better contextualize EM-CoT within this broader research lineage.
>
>
> ---
> ### W2: Clearer analysis of efficiency, scalability, and inference overhead.
>
> Compared to simpler VLA baselines, HALO strikes **a favorable balance** between training efficiency, scalability and inference latency. See our responses to **Reviewer S8VU, W1 & Q1** and **W3 & Q3** for detailed discussions on inference overhead and training efficiency, respectively.
>
>
>
> [1] Do as i can, not as i say: Grounding language in robotic affordances. arXiv:2204.01691
>
> [2] ReAct: Synergizing Reasoning and Acting in Language Models. ICLR 2023
>
> [3] DWIM: Towards Tool-aware Visual Reasoning via Discrepancy-aware Workflow Generation & Instruct-Masking Tuning. ICCV 2025
>
> [4] Code as policies: Language model programs for embodied control. ICRA 2023
>
> [5] VoxPoser: Composable 3D Value Maps for Robotic Manipulation with Language Models. CoRL 2023

---

> > ### Author Rebuttal · Reviewer_DxBt · 2026-04-01
> >
> > Thank you for the detailed clarification and for positioning your work within the broader landscape of reasoning-and-acting frameworks. I agree that the cited works share high-level similarities in terms of framework design, as you discussed. In this regard, I believe explicitly highlighting both the similarities and distinctions between your approach and prior work (e.g., emphasizing the direct application to VLA settings and end-to-end integration) in the related work section would further improve clarity and help readers better understand the design space, as well as inspire future directions.
> >
> > I have also reviewed the concerns raised by other reviewers, particularly those regarding efficiency, scalability, and inference-time trade-offs, along with your responses. Based on the current evidence and experimental results, I find the contributions to be meaningful and well-supported.
> >
> > Therefore, **I maintain my original score and recommendation for acceptance**.
> >
> > Finally, I would like to encourage the authors to consider releasing their code in the future, as this would greatly facilitate broader adoption and further research in this area.

---

> > > ### Author Response · Authors · 2026-04-01
> > >
> > > Thank you for your recognition and encouragement, as well as for your positive evaluation of our work and recommendation for acceptance. We are currently organizing the code and checkpoints, and plan to release them within one month to facilitate future research and broader adoption.

---

### Decision · Program_Chairs · 2026-04-30

**Decision:**

Accept (regular)

**Comment:**

The paper proposes HALO, a unified VLA model that enables embodied multimodal chain-of-thought reasoning. It received four reviews, with major concerns centered on system complexity and efficiency analysis (DxBt, 6zLc, WLLY, S8VU), clearer discussion and comparison with prior reasoning-based methods (DxBt), the need for additional ablation studies (6zLc, WLLY), and robustness to noisy COT (S8VU).

The rebuttal provided clarifications and additional experimental results that address these concerns. The final ratings are 5, 5, 3, and 3. Among the two more negative reviews, WLLY raised concerns about the lack of in-depth analysis, which were addressed in the rebuttal and deemed satisfactory by the AC. The other reviewer (S8VU) did not provide concrete justification for rejection.

Given the strengths of the paper, the AC recommends acceptance. The authors should include the new analysis and results in the revised paper.